# Adsorption Behavior and Relative Distribution of Cd^2+^ Adsorption Mechanisms by the Magnetic and Nonmagnetic Biochars Derived from Chicken Manure

**DOI:** 10.3390/ijerph17051602

**Published:** 2020-03-02

**Authors:** Fei Huang, Lu Zhang, Ren-Ren Wu, Si-Ming Zhang, Rong-Bo Xiao

**Affiliations:** 1Guangdong Industrial Contaminated Site Remediation Technology and Equipment Engineering Research Center, School of Environmental Science and Engineering, Guangdong University of Technology, Guangzhou 510006, China; feihuang2011@163.com; 2College of Natural Resources and Environment, South China Agricultural University, Guangzhou 510642, China; xuml0916@163.com (L.Z.); bigchuang502@163.com (S.-M.Z.); 3Guangdong Key Laboratory of Water and Air Pollution Control, South China Institute of Environmental Sciences, Ministry of Ecology and Environment, Guangzhou 510530, China; wurenren@scies.org

**Keywords:** heavy metal, magnetic biochar, adsorption mechanism, magnetic separation, wastewater treatment

## Abstract

The present study investigated the adsorption of Cd^2+^ by nonmagnetic and magnetic biochars (CMB and M-CMB) derived from chicken manure, respectively. The adsorption characteristics were investigated as a function of initial pH, contact time, initial Cd^2+^ concentration and magnetic separation. Adsorption process of both biochars were better described by Pseudo-second-order kinetic equation and Freundlich isotherm model, which were spontaneous and endothermic in nature. It was found that maximum capacities were 60.69 and 41.07 mg/g obtained at the initial Cd^2+^ concentration of 180 mg/L for CMB and M-CMB, and the turbidity of adsorption-treated solution was reduced from 244.3 to 11.3 NTU after magnetic separation of 0.5 min. These indicated that M-CMB had lower adsorption capacity of Cd^2+^ than CMB, though it was successfully separated from the treated solutions. Furthermore, both biochars before and after adsorption were analyzed by SEM-EDS, XRD and FTIR. Adsorption mechanisms mainly included precipitation, ion-exchange, complexation and Cπ-coordination, in which precipitation and ion-exchange dominated the adsorption process by CMB, while in M-CMB, precipitation was always predominant mechanism, followed by ion-exchange. The two other mechanisms of complexation and Cπ-coordination were trivial in both biochars, jointly contributing 7.21% for CMB and 5.05% for M-CMB to total adsorption. The findings deepen our understanding of the mechanisms governing the adsorption process, which are also important for future practical applications in the removal of heavy metals from wastewater by the biochars.

## 1. Introduction

Cadmium (Cd) is one of the most toxic heavy metal pollutants in wastewater, since it accumulates in living organisms and passes through the food chain into human organs, causing serious toxicity even at low concentration of 0.001–0.1 mg/L [1]. Significant quantities of Cd are introduced into the environment by anthropogenic activities, including manufacturing of nickel-cadmium batteries, synthetic pigment production, metal coatings, stabilizers in plastic products, and incineration of solid wastes, in which the worldwide production of Cd in 2005 was estimated to be 20,000 metric tons [2]. Accordingly, the adsorption using agricultural waste materials is one of the most popular and effective processes for the removal of heavy metals from wastewater. Recently, the attention has been diverted towards the biochar that is obtained from the pyrolysis process of biomass including crop residues, manure and sewage sludge under low temperature (<700 °C) and oxygen-limited environment [3].

Among the biochars, magnetic biochar have been extensively used for heavy metal adsorption, since it could remove Cd^2+^ from aqueous solution to some extent and is easily separated from the wastewater [4]. For example, Li et al. [5] reported the synthesis of magnetic biochar by direct heating of siderite and rice husk under N_2_ condition, and indicated that the adsorption capacity of U(VI) was significantly enhanced after magnetization. While Son et al. [6] developed an engineered magnetic biochar by pyrolyzing waste marine macro-alga, and found that the adsorption efficiency of heavy metal was partially reduced by magnetic biochar. These investigations and other studies [7,8,9,10] showed that the adsorption capacity of heavy metal by magnetic biochar, might be higher or lower than to that of nonmagnetic biochar. Therefore, it is necessary to evaluate the effect of magnetism loading on biochar in terms of the cationic heavy metal adsorption capacity, before a decision for the potential use of magnetic biochar as an industrial adsorbent is made.

The mechanisms responsible for Cd^2+^ adsorption by the biochars include: (i) precipitation with minerals (such as PO_4_^3−^, CO_3_^2−^, SiO_3_^2−^) [11,12]; (ii) exchange with cations (e.g., K^+^, Ca^2+^, Na^+^, Mg^2+^) [13,14]; (iii) surface complexation with oxygen-containing functional groups (e.g., −OH, −COOH, −R−OH) [15,16]; and (iv) coordination with Cπ electrons (e.g., C = C, C ≡ C) [13,17]. Among these mechanisms, Cui et al. [18] reported that the precipitation with minerals was the primary mechanism for Cd^2+^ adsorption on high-temperature biochars derived from *Canna indica*. Zhang et al. [12] showed that the Cd^2+^ adsorption primarily involved coordination with Cπ electrons in the biochar derived from *Phyllostachys pubescens*. Likewise, Wang et al. [16] indicated that Cπ-coordination was the predominant mechanism for Cd^2+^ adsorption by the biochars derived from bamboo, straw and pig manure. Despite the previous work, there is still a lack of existing literature on the comparison between magnetic and nonmagnetic biochar, focusing on the relative contribution of different mechanisms to total adsorption.

To summarize, the main objectives of the present study were to examine the adsorption characterization of Cd^2+^ by magnetic and nonmagnetic biochars derived from chicken manure, respectively, and to understand the contributions of each mechanism to total adsorption on a qualitative and quantitative basis by means of SEM-EDS, XRD and FTIR analysis.

## 2. Materials and Methods

### 2.1. Biochar Preparation and Characteristics

Chicken manure was collected from the farmland at South China Agricultural University, Guangzhou, China. The raw material was dried at 80 °C for 48 h until moisture was evaporated completely, and then sieved to <2.0 mm particles by a stainless grinding machine. For the synthesis of magnetic biochar (M-CMB), FeCl_3_·6H_2_O (20.0 g) and FeSO_4_·7H_2_O (11.1 g) was added to the beaker with 600 mL of deionized water and stirred until they were dissolved completely, after which the powdered biomass was mixed with the solutions and stirred at room temperature (25 ± 2 °C) for 20 min. Thereafter, 10 M-NaOH (aqueous) was added drop wise into the mixed suspension until the pH reached to 10–11. After being stirred for 30 min, the suspension was slowly pyrolyzed in a muffle furnace under N_2_ atmosphere at 600 °C for 4 h according to the methods of Mohan et al. [19]. As a control, nonmagnetic biochar sample (CMB) was prepared under the same pyrolysis conditions, as previously described by Huang et al. [20].

The pH of biochar in water solution was determined at 1:20 (*w*/*v*) ratio after stirring for 1 h, and the content of ash was measured by heating at 800 °C for 4 h in a muffle furnace. To determine the point of zero charge (pHpzc), the zeta potential at various pH values was examined by a Zeta Meter (Malvern, Nano ZS90, UK). The BET surface area was measured by N_2_ adsorption at liquid nitrogen temperature using a Gemini 2360 Micromeritics surface area analyzer and Brunauer-Emmett-Teller equation, and the content of C, H, O, N and S was determined by an elemental analyzer (Flash EA1112, Thermo Finnigan, Italy). The minerals content of K, Ca, Na and Mg were determined by a flame atomic absorption spectrophotometer (FAAS) (M6, Termo Elemental, Waltham, MA, USA), respectively. The content of Fe and Cd in digestion solution were measured by inductively coupled plasma mass spectrometry (ICP-MS, Thermo Fisher Scientific, Waltham, MA, USA). The concentrations of water-soluble PO_4_^3−^ and CO_3_^2−^ were measured by ion chromatography (IC1000, Dionex Co., Ltd. Sunnyvale, CA, USA). The magnetic properties of biochar were determined by a comprehensive physical measurement system (PPMS-9, Quantum Design, San Diego, CA, USA).

### 2.2. Adsorption Experiments

Metal stock solutions of 1000 mg/L were prepared by dissolving CdCl_2_·2.5H_2_O (GR, guaranteed reagent) in double distilled water, and then diluted to the desired concentration prior to the adsorption experiments. Batch adsorption were performed by adding 0.03 g biochar samples to 30 mL synthetic solutions containing a known initial Cd^2+^ concentration at room temperature (25 ± 2 °C), and agitated at a speed of 150 rpm. The concentrations of ions including Na^+^ and Cl^−^ were lower than the detection limits, and the initial pH of mixed solutions was adjusted to 6.01 ± 0.2 by 0.1 M NaOH or HCl. The effect of pH on the adsorption was investigated in range of 2.0–8.0, and carried out by mixing 1 g/L adsorbents at the initial Cd^2+^ concentration of 100 mg/L. For adsorption kinetics, biochar samples were mixed with synthetic water containing 20, 50, and 100 mg Cd^2+^/L at pH 6.0, respectively, and the residual Cd^2+^ concentration was determined at various time intervals up to 360 min. After adsorption equilibrium, the final suspensions were centrifuged (8000 rpm, 10 min) and then filtered (0.22 μm Millipore filter), the supernatant was thus prepared to the measurement of Cd^2+^. Likewise, adsorption isotherm experiments were carried out at initial Cd^2+^ concentrations in the range of 10–180 mg/L under the temperature of 293, 303 and 313 K, respectively. The magnetic separation of biochar particles from aqueous solution was investigated using a magnetic field of 0.5 T, according to the methods of Wang et al. [21].

Lagergren’s pseudo-first-order equation (Equation (1)), and pseudo-second-order equation (Equation (2)) were used to fit the adsorption kinetics process. Langmuir (Equation (3)), and Freundlich (Equation (4)) isotherm equations were applied to model the experimental data. Adsorption thermodynamic (Equations (5) and (6)) was established by fixing free energy (∆G^0^), enthalpy (∆H^0^) and entropy (∆S^0^), according to the methods described by Milonjić [22].
(1)$$q_{t}=q_{e}\left(1-e^{-k_{1}t}\right)$$
(2)$$q_{t}=\frac{k_{2}q_{e}^{2}t}{1+k_{2}q_{e}t}$$
(3)$$q_{e}=\frac{q_{\max}K_{L}C_{e}}{1+K_{L}C_{e}}$$
(4)qe=KFCe1n
(5)$$\Delta G^{0}=-\mathrm{RTln}\left(\rho_{\omega }K_{D}\right)$$
(6)$$\ln\left(\rho_{\omega }K_{D}\right)=\frac{\Delta S^{0}}{R}-\frac{\Delta H^{0}}{\mathrm{RT}}$$
where q_e_ and q_t_ are the adsorption capacity at equilibrium and time t (mg/g), respectively. k_1_ (l/min), and k_2_ (g/mg min) are the rate constants corresponding to the respective kinetic model. Ce (mg/L) is the residual Cd^2+^ concentration at equilibrium, q_max_ (mg/g) is the maximum adsorption capacity, K_L_ (L/mg), K_F_ (L/g) and n are the rate constants corresponding to the respective isotherm model. ∆G^0^ (kJ/mol), ∆S^0^ (kJ/mol/K), and ∆H^0^ (kJ/mol) represent the changes of Gibb’s free energy, entropy and enthalpy, respectively. K_D_ (L/g) is the thermodynamic equilibrium constant (Langmuir isotherm constant) and ρ_w_ (g/L) is the water density. R (8.314 J/K/mol) is the gas constant and T (K) is absolute temperature. In the case when K_D_ is given in L/mg, the constants of ρ_w_K_D_ can be easily recalculated to become dimensionless by multiplying it by 1,000,000 Milonjić [22].

### 2.3. The Contribution of Different Adsorption Mechanisms

Both biochars (CMB and M-CMB) were demineralized by rinsing with 1 M HCl and distilled water until the pH of liquid became constant, then untreated and demineralized biochars (0.03 g) were added into 30 mL solutions with a known initial Cd^2+^ concentration (100 mg/L) and without Cd^2+^, respectively. After adsorption, the reduced amount of Cd^2+^ by the biochars before and after demineralization could be considered as the contribution of these removed minerals, because most minerals in the biochars were removed by the acid dipping procedure, but the surface oxygen-containing functional groups were not altered [12,23]. After the adsorption of untreated biochars (CMB and M-CMB), the cations including K^+^, Na^+^, Ca^2+^ and Mg^2+^ in the solution with and without Cd^2+^ were determined by ICP-MS. After that, a drop of pH before and after Cd^2+^ adsorption on both demineralized biochars was merely from the complexation with the oxygen-containing functional groups. In addition, the biochars loaded with and without Cd^2+^ were prepared for chemical analysis using scanning electron microscopy with energy dispersive X-ray spectroscopy (SEM-EDS, Hitachi, Japan), X-ray diffractometer (XRD-6000, Shimadzu, Japan) and fourier transform infrared spectroscopy (FTIR, PerkinElmer 2000, Waltham, WA, USA).

The adsorption capacity resulting from precipitation (Q_pre_), ion-exchange (Q_exc_), complexation (Q_com_), Cπ-coordination (Q_cπ_) and total adsorption (Q_ct_), were calculated by the modified methods of Gao et al. [1] and Wang et al. [23], respectively. The corresponding contribution percentage of each mechanism to total adsorption was determined by the Q_pre_/Q_ct_, Q_exc_/Q_ct_, Q_com_/Q_ct_ and Q_cπ_/Q_ct_ ratio.

(i) The adsorption capacity resulted from precipitation was calculated by the difference in the adsorption amount of Cd^2+^ between on untreated and demineralized biochars.
(7)$$Q_{\mathrm{pre}}=Q_{\mathrm{ct}}-Q_{\mathrm{ca}}*Y$$
where Q_ca_ is the adsorption capacity of demineralized biochars (mg/g), Y is the yield of demineralized biochar from original biochar.

(ii) The adsorption capacity attributed to ion-exchange was obtained by subtracting cations released in the 0 mg Cd^2+^/L solution as background concentration from total released cations in the100 mg Cd^2+^/L solution.
(8)$$Q_{\mathrm{exc}}=Q_{k}+Q_{\mathrm{Ca}}+Q_{\mathrm{Na}}+Q_{\mathrm{Mg}}$$
where Q_k_, Q_Ca_, Q_Na_, Q_Mg_ are the adsorbed Cd^2+^ by net release amount of K^+^, Ca^2+^, Na^+^ and Mg^2+^ during the adsorption process, respectively (mg/g).

(iii) The adsorption capacity resulted from complexation was estimated by the difference in pH change before and after adsorption on the demineralized biochars.
(9)$$Q_{\mathrm{com}}=Q_{\mathrm{pH}}*Y$$
where Q_pH_ is the adsorbed Cd^2+^ by complexation with oxygen-containing function groups on demineralized biochars (mg/g).

(iv) The adsorption capacity resulted from Cπ-coordination was calculated by subtracting the sum of Q_pre_, Q_exc_ and Q_com_ from total adsorption capacity.
(10)$$Q_{c\pi }=Q_{\mathrm{ct}}-Q_{\mathrm{pre}}-Q_{\mathrm{exc}}-Q_{\mathrm{com}}$$

### 2.4. Statistical Analysis

All adsorption experiments were conducted in triplicate, and the standard deviation was calculated by descriptive statistics. Data were statistically analyzed by one-way analysis of variance (ANOVA) at a 0.05 probability level using SPSS 18.0.

## 3. Results and Discussion

### 3.1. Biochar Properties

Both biochars were alkaline with high pH values (Table 1, 11.43 for CMB and 11.55 for M-CMB), suggesting that it would potentially remove heavy metals from acid mine drainage [24]. This was correlated to ash content, since the content of ash showed a positive correlation with the alkalinity of bicohar, as observed by many researchers [20,25,26]. High content of ash was also partly reflected by the SEM observations exhibiting many impurities on the surface of biochars (Figure 1). Moreover, the ash contained a large amount of alkaline mineral elements such as K, Ca, Na and Mg, as demonstrated by EDS and XRD results, which might begin to separate from the organic matrix during the pyrolysis process at high temperature (>300 °C), resulting in the increase of pH to strong basicity [25]. In addition, the surface area decreased from 25.56 m^2^/g in CMB to 5.44 m^2^/g in M-CMB, which could be attributed to the synthesis of magnetic biochar, causing many micropores were covered by the ash of M-CMB. Similar phenomenon was obtained in the previous work of some researchers reporting the reduction in the surface area after magnetization, because the Fe-containing particles reduced the surface area by filling micropores [6,10,27].

Compared with CMB, the elemental content of M-CMB including C, H, O, and N decreased obviously, resulting in the increase of atomic ratio of H/C and O/C (Table 1). This suggested the increase in abundance of oxygen-containing functional groups such as hydroxyl and carboxylic, which might be responsible for the Cd^2+^ adsorption by surface complexation or Cπ-coordination [15,28]. Additionally, CMB contained higher content of alkaline metal cations and soluble anions compared to M-CMB, which could play an important role in the Cd^2+^ adsorption by ion-exchange and precipitation, respectively [29,30].

After the magnetization, M-CMB showed ferromagnetic properties with the saturation magnetization (Ms) of 64.96 emu/g (Appendix A), which was higher than that of pure Fe_3_O_4_ materials (Ms = 58.94 emu/g) [21]. Coincidentally, the SEM-EDS results implied that the iron particles might aggregate to the surface of magnetic biochar (Figure 1B), which was indicated by the FTIR spectra observing the stretching vibrations of Fe-O (around 565 cm^−1^) [31], and further confirmed by XRD pattern implying the formation of Fe_3_O_4_. These suggested that M-CMB presented a superior magnetic response, because of massive coverage of biochar surface by iron particles.

### 3.2. Adsorption Characteristics

#### 3.2.1. pH Adsorption Edge

Both adsorption capacities increased with increasing pH, and began to decline after reaching maximum adsorption at approximately pH 6.0, at which the maximum adsorption capacities were 51.37 mg/g for CMB and 28.30 mg/g for M-CMB, respectively (Figure 2A). The pH_pzc_ was calculated to be 2.18 and 2.44 for CMB and M-CMB, respectively (Figure 2B). When pH < pH_pzc_, the surface charge of biochars became positive, causing electrostatic repulsion to positively charged Cd^2+^, thus low level of adsorption was observed at pH 2.0 for both biochars. While pH > pH_pzc_, the surface charge of biochars became negative, resulting in the significant increase of adsorption capacity from pH 2.0 to 6.0 [16]. Note that CMB showed higher adsorption capacities of Cd^2+^ than M-CMB, which may be due to the greater amount of negative charge, or larger of specific surface area of CMB compared to M-CMB under the pH range studied (Figure 2A,B). A similar phenomenon that the biochars have lower pH_IEP_ but higher adsorption capacity, was also reported in many previous works reporting that electrostatic interaction was likely the driving force for the Cd^2+^ adsorption onto biochar surfaces [1,32,33].

Compared with the initial pH, significant increase of equilibrium pH was observed for both biochars (Figure 2C), which was resulted from the addition of alkaline biochars to neutralize the solution acidity. Compared with the system without Cd^2+^, equilibrium pH was lower when Cd^2+^ was adsorbed onto biochars (Figure 2C,D). This reduction could be attributed to surface complexation between Cd^2+^ and oxygen-containing functional groups such as −COOH and −OH, this was accompanied with the release H^+^ into the solution, resulting in the decrease of equilibrium pH after adsorption [16,34]. Another explanation may be the formation of the Cd precipitate with PO_4_^3−^ and CO_3_^2−^, reducing equilibrium pH after adsorption [18,20]. These observations suggested that electrostatic ion-exchange, complexation, and precipitation were involved in the adsorption process.

#### 3.2.2. Adsorption Kinetics

Both adsorption generally exhibited initial high removal efficiency within 30 min, and then reached the equilibrium within almost 150 min (Figure 3). The adsorption capacities of CMB were 15.56 mg/g, 28.20 mg/g, and 37.97 mg/g within 150 min at initial Cd^2+^ concentrations of 20, 50 and 100 mg/L, accounting for 97%, 99%, and 97% to total capacity, respectively. In contrast, it seemed that the adsorption of M-CMB took longer to reach equilibrium, especially at the initial Cd^2+^ concentrations of 50 mg/L (Figure 3B). The adsorption capacities of M-CMB were much lower than those of CMB at the experimental conditions.

Adsorption kinetics were assessed by pseudo-first-order and pseudo-second-order kinetic model, respectively (Figure 3). For both biochars, all the regression coefficients (R^2^) of pseudo-second-order kinetic model were above 0.95, which were much larger than those of pseudo-first-order kinetic model (Appendix A). Moreover, the adsorption capacities at equilibrium (Q_e,cal_) calculated from pseudo-second-order kinetic model were very close to the experimental data (Q_e, exp_). These indicated the adsorption followed the pseudo-second-order mechanism and corresponded to a chemisorption process [35]. In this regard, many studies indicated that the pseudo-second order model could predict almost all adsorption kinetic processes between metals and biomaterials, because of the basic on the adsorption capacity of solid adsorbents that forecasted the adsorption behavior over the entire process [36,37,38].

#### 3.2.3. Adsorption Isotherms

Adsorption capacities increased rapidly with increasing initial Cd^2+^ concentration for both biochars, and leveled off approximately when the initial metal concentrations were above 120 mg/L in all cases (Figure 4A–C). These could be explained by a stronger driving force with increasing Cd^2+^ concentrations, leading to higher probability of collision between Cd^2+^ and biochars [39]. The maximum capacities were 42.44, 47.45 and 60.69 mg/g for CMB at the temperature of 293 K, 303 K, and 313 K, respectively. These were higher than those of M-CMB, with the respective maximum values of 29.32, 29.39, and 41.07 mg/g. Such difference in the adsorption capacities of both biochars was consistent with the results from adsorption kinetics (Figure 3) and pH adsorption edge (Figure 2), which were correlated with the different physico-chemical characteristics of both biochars.

The experimental data were better fitted with the Freundlich isotherm than Langmuir isotherm under the temperature of 293 K, 303 K, and 313 K, respectively (Figure 4D–F). For both biochars, all the R^2^ values of Freundlich model were greater than 0.95 (0.95–0.99), which were much higher than those of Langmuir model (0.67–0.90). Meanwhile, all the 1/n values were observed in the ranges of 0.13–0.45 (Appendix A), which satisfied the favorable adsorption condition (0 < R_L_ < 1) under the concentration ranges studied [35]. These indicated the chemisorption of Cd^2+^ onto both biochars had a great heterogeneity of adsorption affinity [40]. These results were in line with published data, describing that heavy metal adsorption were well fitted to Freundlich model by the magnetic biochars derived from rice husk [41], *loofah sponges* [42], and *chlorella vulgaris* [43].

#### 3.2.4. Adsorption Thermodynamic

The adsorption process was spontaneous and endothermic in nature, with negative values of free energy (∆G^0^), positive values of enthalpy (∆H^0^) and entropy (∆S^0^) (Table 2). For both biochars, the negative values of ∆G^0^ suggested that the spontaneous and feasible nature of adsorption process at all the studied temperatures. The negative values of ∆G^0^ decreased as the rising temperature in CMB, implying the driving force to the adsorption increased with increasing temperature [44]. Meanwhile, the positive values of ∆H^0^ highlighted the endothermic nature of adsorption by both biochars, and the positive values of ∆S^0^ indicated randomness slightly increased during metal chemisopriotn from solution onto the surface of biochar [8,45].

#### 3.2.5. Magnetic Separation of M-CMB

The turbidity of solution decreased from 244.3 to 11.3 NTU after magnetic separation within 0.5 min, whereas that of the one without magnetic field was 179.7 NTU (Figure 5). Moreover, the turbidity of solution kept relatively high (38.3 NTU) only by gravity after 50 min, while the magnetic separation rate was very rapid, owing to strong magnetism of M-CMB (Ms = 64.96 emu/g, Appendix A). These suggested that magnetic biochar particles could be successfully separated from the treated-solution with the help of external magnet.

Combined with adsorption experimental data, it suggested that CMB had higher adsorption capacity of Cd^2+^ than that of M-CMB, this might be largely attributed to higher content of soluble ions for CMB, such as K^+^, Ca^2+^, Na^+^, Mg^2+^, CO_3_^2−^ and PO_4_^3−^ (Table 1), which was beneficial to the adsorption of Cd^2+^ through mixed mechanisms involving surface complexation, precipitation, ion-exchange, and Cπ-coordination. Similarly, the finding that the magnetization of biochar significantly reduced the adsorption efficiency, were also reported in the previous works, attributing it to the active sites of adsorbent being occupied by iron oxide particles [24,46].

### 3.3. Adsorption Mechanisms

#### 3.3.1. Metal Precipitation

To confirm the role of precipitation, the untreated and Cd^2+^-loaded biochars (150 mg/L) were scanned by XRD (Figure 6A). These precipitates on Cd-loaded biochars were possibly identified with typical peaks as CdCO_3_, Cd_3_(PO_4_)_2_, CdSiO_3_, Cd(OH)_2_, CdS, CdP_2_, and CdFe_2_O_4_, although their presence cannot prove definitely. Coincidentally, some white granular crystals were visible on the biochars surface from the SEM observations (Figure 1), and their elemental composition mainly included Cd, C, O, P, S, Fe and Si, as indicated by EDS spectrum. This might partly support the occurrence of surface precipitation with minerals, or surface complexation with oxygen-containing functional groups [47]. The unfamiliar reduction state of P was detected as CdP_2_ after adsorption, which might be resulted from the release of CH_4_, H_2_ and CO during pyrolysis [48]. Among these precipitates, such as CdCO_3_, Cd_3_(PO_4_)_2_, CdSiO_3_, Cd(OH)_2_ and CdS, were probably due to high concentrations of CO_3_^2−^, PO_4_^3−^, S and Si content in the original biochars (Figure 1 and Table 1), which were also observed in the previous studies on the heavy metal adsorption by the biochars [1,49,50]. In particular, the precipitates of CdFe_2_O_4_ were detected on the surface of M-CMB after adsorption, which could be attributed to the presence of Fe_3_O_4_ particles, generating adsorption sites for metal ions [6,10]. Generally, the intensities of XRD peak were strong in both biochars, suggesting the precipitation played an important role in total adsorption.

#### 3.3.2. Surface Complexation

FTIR analysis was applied to characterize the changes of functional groups before and after adsorption (Figure 6B). For M-CMB, the peaks at 3779 cm^−1^ and 3405 cm^−1^ representing O-H vibrations of hydroxyl groups, were disappeared after interaction with Cd^2+^ [24]. The peak at 1626 cm^−1^ was attributed to C = O stretching vibration of carboxyl, were significantly decreased after adsorption [47]. The band at 1414 cm^−1^ was assigned to CO_3_^2−^ and the peak at 1045 cm^−1^ was due to P-O stretching vibrations of PO_4_^3−^ [51], these were changed after adsorption, supporting the occurrence of the metal-carbonate and metal-phosphate precipitates, respectively (Figure 6A). Note that the bands between 880–780 cm^−1^, 1100–1000 cm^−1^, and 700–400 cm^−1^ were characteristic of Si-O-Si symmetric stretching, asymmetric stretching, and bending, respectively [52]. Considering that low content of Si in the original biochar (Figure 1), it would be more accurate to state that Si element probably participates in the adsorption by forming CdSiO_3_, as partly supported by XRD analysis (Figure 6A). In particular, the peak at 446 cm^−1^ could be assigned to the stretching vibrations of Fe-O, testifying the formation of iron oxides [21]. In contrast, the bands of CMB at 2983 cm^−1^ (-CH stretching), 2513 cm^−1^ (CO_3_^2−^), 1799 cm^−1^, 1037 cm^−1^ (P-O stretching vibrations of PO_4_^3−^), 874 cm^−1^ (Si-O-Si symmetric stretching), and 567 cm^−1^ (Si-O-Si bending) were all increased slightly after adsorption. These suggested that oxygen-containing functional groups such as -OH, -COOH, as well as C-H aromatic group might be involved in the adsorption. However, the intensities of FTIR spectra in both biochars changed slightly within 1–2 units, implying the complexation was an insignificant mechanism in total adsorption. Indeed, the adsorption capacity resulting from this mechanism were 2.13 mg/g and 1.26 mg/g for CMB and M-CMB, respectively (Table 3).

#### 3.3.3. Cπ-Coordination

The cyclic aromatic π-system formed in the biochars could function as the π-donor in the Cd^2+^ adsorption [17]. According to the FTIR spectra (Figure 6B), the peaks between 900–700 cm^−1^ and 700–400 cm^−1^ representing the aromatic C-H groups changed slightly after adsorption, suggesting aromatic functional groups such as γ-CH of furan and β–ring of pyridine might be involved in the adsorption by both biochars [30]. These heterocyclic compounds were a weak cation-π binder and easily bind with Cd^2+^ by donating π electrons [53], it was thus reasonable to believe that the coordination with Cπ electrons would occur in the adsorption. Similar results were reported by Harvey et al. [13] and Machida et al. [54], who found that the heavy metal adsorption by biochars were mainly attributed to the coordination with delocalized π electrons from the functional groups such as γ-CH and C=C. The formation of Cd-Cπ bonds between Cd^2+^ and delocalized lone-pair π electrons could be described by the following reaction:


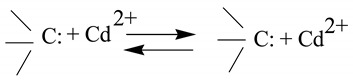
(11)

#### 3.3.4. Ion-Exchange

Ion-exchange was considered as the dominant mechanism for heavy metal adsorption by the biochars [13,30]. In parallel with the Cd^2+^ adsorption, significant amount of K^+^, Ca^2+^, Na^+^ and Mg^2+^ were released into the solution, equivalent to 22.75 and 8.01 mg Cd^2+^/g (Table 4), accounting for 47.62% and 23.90% of total adsorption by CMB and M-CMB, respectively (Figure 7). These indicated that cation exchange played an important role in total adsorption. Moreover, the dominance of K^+^ and Ca^2+^ among these released cations was observed in CMB, this was seemed fairly comparable to the studies of Flores-Cano et al. [14] and Chen et al. [55], who proposed that Cd^2+^ substitution for Ca^2+^ was a major mechanism in the adsorption.

### 3.4. Relative Distribution of Adsorption Mechanisms

The precipitation formed the biggest fraction in M-CMB, because the Q_pre_ was 23.83 mg/g with relatively high contribution of 71.06% to total adsorption (Figure 7), highlighting that the precipitation was the dominant mechanism in total adsorption. While in CMB, the Q_pre_ was 21.57 mg/g, whose contribution proportion was 45.18% in total adsorption. Similar to the precipitation, ion-exchange produced the largest contribution to total adsorption for CMB (47.61%), though its importance was decreased slightly in M-CMB (23.89%). This situation seemed fairly comparable to the finding, in which the heavy metal adsorption by dairy-manure biochar was mainly attributed to the precipitation with PO_4_^3−^ and CO_3_^2−^ [51], and the adsorption by rice-straw biochar was closely related to the exchangeable cations [50], respectively. Thus, the precipitation and ion-exchange dominated the adsorption for both biochars, jointly contributing 92.79% in CMB and 94.95% in M-CMB to total adsorption. On the other hand, complexation and Cπ-coordination were comparably tiny among these mechanisms, since both Q_com_ and Q_cπ_ values were relatively low. The values of Q_com_ were 2.13 and 1.26 mg/g for CMB and M-CMB, with their corresponding contribution proportions were 4.46% and 3.76%, respectively. The Q_cπ_ values were 1.31 mg/g for CMB and 0.43 mg/g for M-CMB, whose contribution proportions were 2.75% and 1.29%, respectively.

Based on the present results, magnetic biochar like M-CMB should be further modified to enhance its adsorption ability for heavy metal ions. As a result, a variety of approaches would be applied to improve the adsorption ability of magnetic biochar, such as oxidation [56], amination [46] and loading of metal nanoparticles [57]. Among these methods, more efforts should be paid to explore the effect of new functional groups-impregnated magnetic biochar after modification, since there is a great potential in the enhancement of adsorption capacity for M-CMB, mainly by improving complexation and Cπ-coordination.

## 4. Conclusions

Compared with the M-CMB, CMB have greater adsorption capacities, faster adsorption kinetics, but have difficulties in separating from the solution after adsorption. For both biochars, the adsorption process followed chemisorption mechanism, occurring on the heterogeneous surface at tested conditions. Among the four mechanisms, the precipitation and ion-exchange dominated the adsorption, in combination contributing 92.79% for CMB and 94.95% for M-CMB, but complexation and coordination were comparably tiny in total adsorption. These could help to improve the application of biochar in environmental remediation for heavy metal. However, the research regarding the enhancement of adsorption capacity by magnetic biochar should be further investigated in future.

## Figures and Tables

**Figure 1 ijerph-17-01602-f001:**
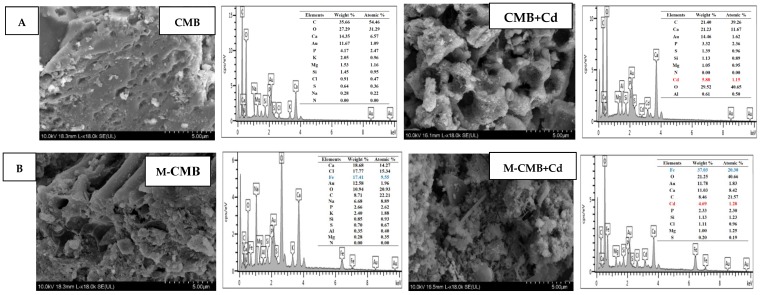
SEM images (left) and corresponding EDS spectra (right) of CMB (**A**) and M-CMB (**B**) before and after adsorption for Cd^2+^.

**Figure 2 ijerph-17-01602-f002:**
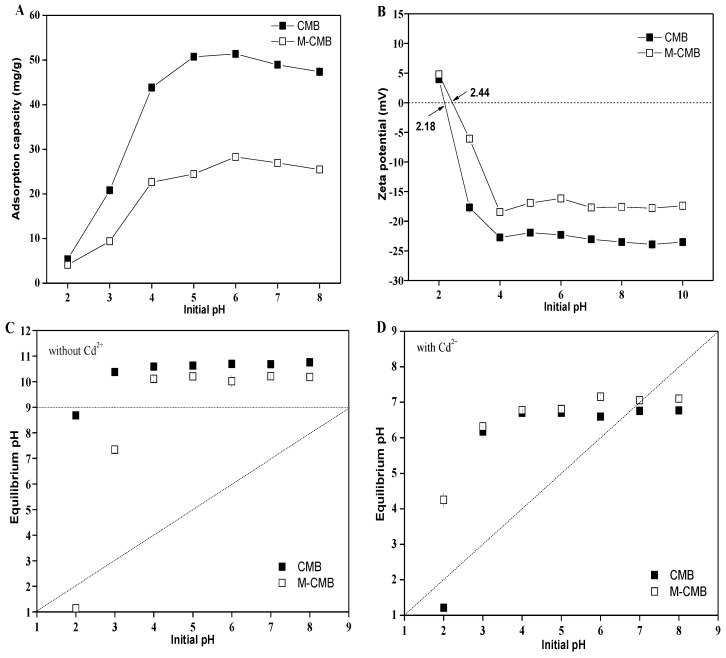
Effect of pH on the adsorption by both biochars (**A**). Zeta potential of both biochars at different initial pH values (**B**). Equilibrium pH of mixed solution without Cd^2+^ at different initial pH values (**C**). Equilibrium pH of mixed solution with Cd^2+^ at different initial pH values (**D**).

**Figure 3 ijerph-17-01602-f003:**
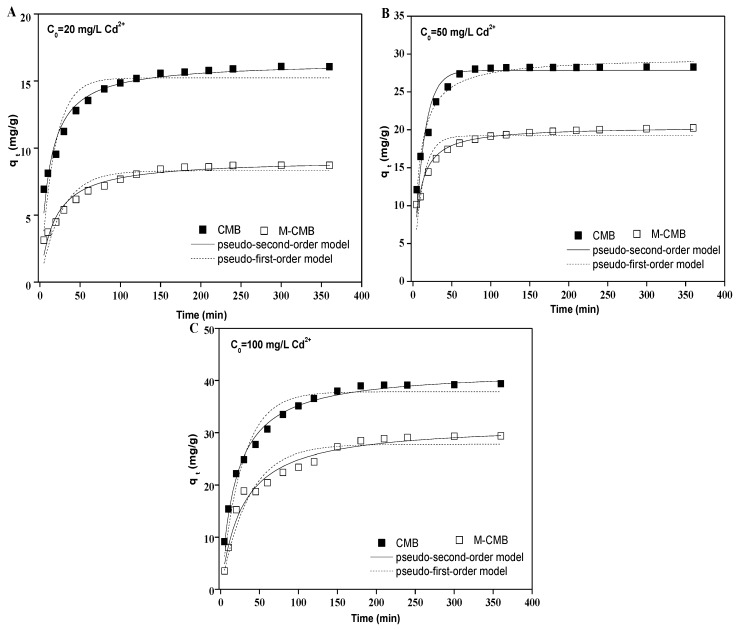
Effect of contact time on the adsorption at initial Cd^2+^ concentrations of 20 (**A**), 50 (**B**) and 100 mg/L (**C**), respectively.

**Figure 4 ijerph-17-01602-f004:**
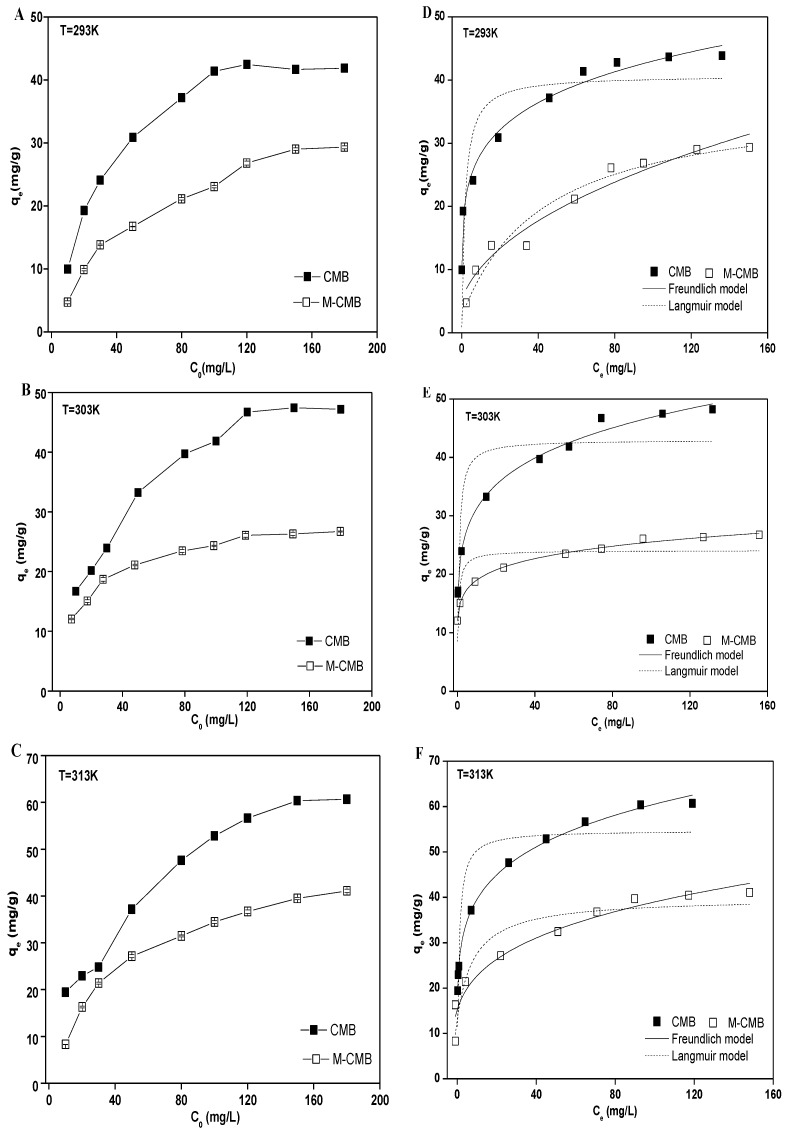
Effect of initial Cd^2+^ concentration on the adsorption under the temperature of 293 (**A**), 303 (**B**) and 313 K (**C**), respectively. Adsorption isotherms of the biochars under the temperature of 293 (**D**), 303 (**E**) and 313 K (**F**), respectively.

**Figure 5 ijerph-17-01602-f005:**
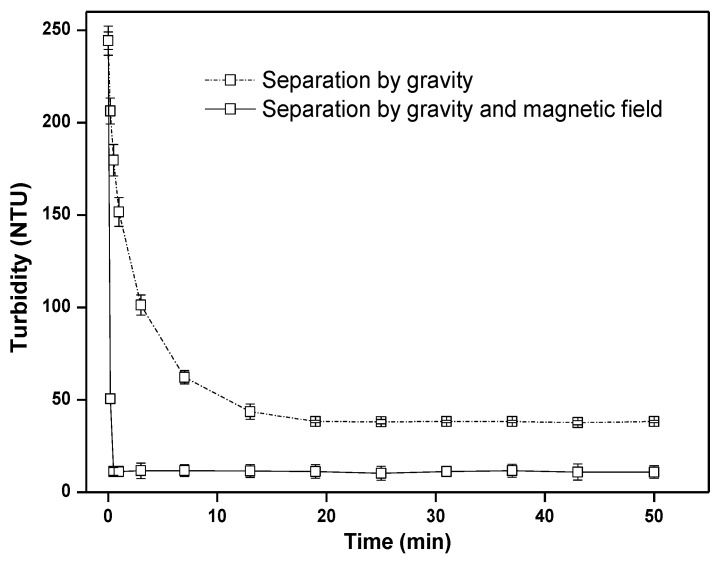
Effect of time on magnetic separation of M-CMB by a magnetic field of 0.5 T.

**Figure 6 ijerph-17-01602-f006:**
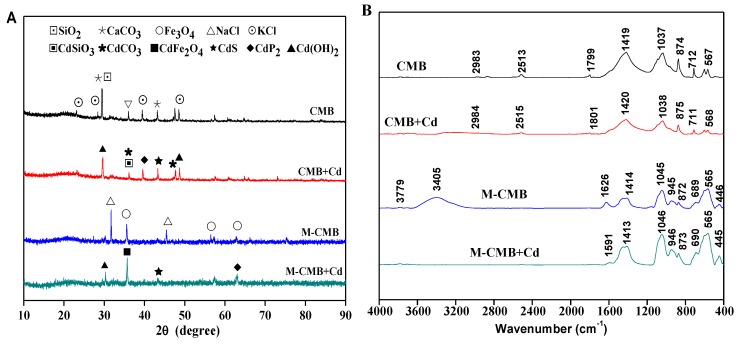
XRD patterns of both biochars before and after Cd^2+^ adsorption (**A**). FTIR spectra of both biochars before and after Cd^2+^ adsorption (**B**).

**Figure 7 ijerph-17-01602-f007:**
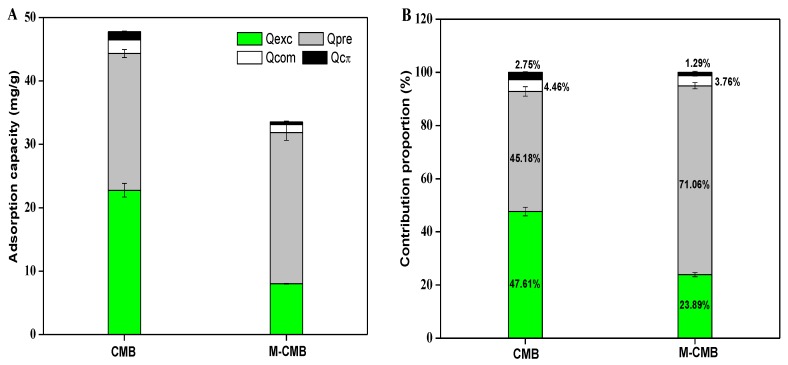
Adsorption capacity of Cd^2+^ adsorption by both biochars (**A**). Contribution percentage of different mechanisms to total adsorption by both biochars (**B**).

**Table 1 ijerph-17-01602-t001:** The main properties of both biochars.

Biochar	pH	pH_pzc_	Ash (%)	S_BET_ (m^2^/g)	Fe (mg/g)	Magnetism (emu/g)	Elemental Content (%)					Atomic Ratio		Soluble Ions Content (mg/g)					
							C	H	O	N	S	H/C	O/C	K^+^	Ca^2+^	Na^+^	Mg^2+^	CO_3_^2−^	PO_4_^3−^
CMB	11.43	2.18	78.64	25.56	—	—	11.35	1.07	23.07	0.89	0.25	1.13	1.52	8.37	0.30	3.34	0.02	1.19	0.03
M-CMB	11.55	2.44	86.64	5.44	10.82	64.96	3.17	0.37	15.93	0.72	0.45	1.40	3.77	5.71	0.02	0.52	0.01	0.63	0.01

S_BET_ is the Brunauer-Emmett-Tell (BET) surface area, m^2^/g. pH_pzc_ is the pH at which the surface charge of biochar is zero.

**Table 2 ijerph-17-01602-t002:** Thermodynamic parameters of adsorption for Cd^2+^ by both biochars.

Biochar	∆G^0^ (kJ mol^−1^)			∆H^0^ (kJ mol^−1^)	∆S^0^ (kJ mol^−1^ K^−1^)
	293 K	303 K	313 K		
CMB	−32.37	−35.63	−36.29	25.54	0.20
M-CMB	−25.11	−36.42	−31.63	73.05	0.34

**Table 3 ijerph-17-01602-t003:** Changes in hydrogen ion concentration of adsorption for Cd^2+^ by both biochars.

Biochar	Initial pH	Final pH	△H^+^(10^−6^ mmol)	Q_e_ (mg/g)
CMB	5.96 ± 0.02	3.87 ± 0.01	134 ± 0.06	2.13 ± 0.02
M-CMB	5.02 ± 0.01	4.36 ± 0.02	34.1 ± 0.18	1.26 ± 0.01

**Table 4 ijerph-17-01602-t004:** The release of K^+^, Ca^2+^, Na^+^ and Mg^2+^ during the Cd^2+^ adsorption by both biochars.

Biochar	The Net Amount of Released Cations (mequiv/g)				Sum	Overall Adsorbed Cd^2+^(mg/g)
	K^+^	Ca^2+^	Na^+^	Mg^2+^		
CMB	14.43 ± 0.26	9.58 ± 0.38	7.10 ± 1.73	2.41 ± 0.07	33.52 ± 1.83	22.75 ± 0.88
M-CMB	7.90 ± 0.13	1.74 ± 0.08	4.11 ± 0.10	0.26 ± 0.02	14.01 ± 0.08	8.01 ± 0.06

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
