# Peer review of "Adsorption Behavior and Relative Distribution of Cd2+ Adsorption Mechanisms by the Magnetic and Nonmagnetic Biochars Derived from Chicken Manure"

_ijerph, 2020, doi:10.3390/ijerph17051602_

Round 1
Reviewer 1 Report
The authors have responded to all my questions and made the appropriate changes in the manuscript. I believe it has a much better flow of information and adequate explanations provided in various sections. I recommend to be considered for publication after my final suggestion.
Line 106: Please provide a description/composition regarding synthetic solution. The manuscript presents a thorough study in Cd removal but no info about the working matrix content, beside metal's addition, is mentioned. In my opinion this is essential.
Author Response
Yes, that is essential for the adsorption experiments. As your said, the occurrence of co-existing ions might affect the adsorption of Cd2+. More details were provided in the new manuscript to better understand, please See the red font in Section 2.2 Lines 104-109.
In the present work, some ions such as Na+ and Cl-, could be commingled by the addition of metal stock solution (CdCl2·2.5H2O, guaranteed reagent GR), or the regulation of pH by NaOH or HCl. According to our measurement, the concentrations of Na+ and Cl- were lower than the detection limits, and the pH values was 7.0±0.2, this might because the synthetic solution was accurately prepared by double distilled water, and the chemical reagent were of GR rather than AR (analytical reagent).
Reviewer 2 Report
Authors responded adequately to all suggestions for the improvement of the quality of the manuscript. However, it is suggested that the term ρw be redefined as dissolution density and not water density (Eq. 6).
Author Response
Authors responded adequately to all suggestions for the improvement of the quality of the manuscript. However, it is suggested that the term ρw be redefined as dissolution density and not water density (Eq. 6).
Thank you very much. The details were provided in the new manuscript, as shown in red font (Lines 130-133).
|
KD (L/g) is the thermodynamic equilibrium constant and w (g/L) is the water density. R (8.314 J/K/mol) is the gas constant and T (K) is absolute temperature.
According to the literature (Milonjić, 2007; Gürses et al., 2014), the constants of wKD for each temperature was converted to dimensionless constants via multiplying by the density of the liquid phase (1 g/ml). In this study, the KD is given in L/mg (Langmuir isotherm constant, L/mg), the constants wKD can be easily recalculated to become dimensionless by multiplying it by 1,000,000.
Milonjić, S.K. A consideration of the correct calculation of thermodynamic parameters of adsorption. Journal of the Serbian Chemical Society, 2007, 72, 1363-1367.
Gürses, A., Hassani, A., KiranÅŸan, M., et al., Removal of methylene blue from aqueous solution using by untreated lignite as potential low-cost adsorbent: kinetic, thermodynamic and equilibrium approach. Journal of Water Process Engineering, 2014, 2, 10-21.

This manuscript is a resubmission of an earlier submission. The following is a list of the peer review reports and author responses from that submission.
Round 1
Reviewer 1 Report
This paper describes and quantifies the contribution of each adsorption mechanism in the removal of Cd2+ by biochar. The designed adsorption study, waste utilization and its monitoring by means of characterization techniques (XRD and FTIR) is original and adequate. However, the paper presents errors in the redaction and in some of the calculations. I suggest that the work be improved taking into account the following guidelines:
Rewrite the beginning of the abstract (lines 16-19). The sentence is too long. You must start by indicating the aim of the research, then what effects were studied, modelling the results and finally the most remarkable result. Detail in which part of the treatment the adsorption of the biochar would be applied (lines 30-31). Add the following keywords: adsorption mechanism; wastewater treatment. Typographical errors: Check superscripts Introduction section. Rewrite the hypothesis. It does not justify the study. (lines 51-53).The introduction is incomplete and poorly written. Rewrite introduction section with the following points:
a. Industrial activities where cadmium-containing effluents are generated
b. Frequent cadmium concentrations in these industrial effluents
c. Treatment alternatives. Emphasize the use of the adsorption process as well as adsorbent materials as an alternative to existing processes in the literature (state of the art).
d. Hypothesis (rearrange with lines 51-54 and 66-68)
e.General characteristics of the biochar used in the study
f. Adsorption mechanisms
g.Objectives of the study
Reviewer 2 Report
The manuscript reports the investigation of Cd(II) adsorption on modified and no-modified biochar, which delivered from chicken manure. However, it should be revised carefully and certainly to be improved. Some comments are listed as follows:
The English level of this manuscript should be polished (minor issues). Page 2, line no. 45: Please rephrase the term “degrees”. It doesn’t read properly in the current form. Page 2, lines no. 87-88: This sentences belong in the results and discussion section. Page 3, lines no. 100-101: The authors mentioned the Cd solution but no information are provided, such as preparation, volume or/and liquid to adsorbent ratio. Please explain. Page 3, line no. 102: The authors mentioned the synthetic water but no information are provided, such as composition. Please explain. Page 5, line no. 179: A comment regarding Fe3O4 coverage should be added, e.g. Fe content (it may also be included in the Table 1), magnetite-biochar ratio etc. Page 7: Please provide the experimental condition in Subsection 3.2.1 (Cdinitial and adsorbent concentration), which should also be mentioned in Materials and Methods section. Page 9, line no. 256: The authors claim that chemisorption is the main mechanism but, according to Table 2, the ΔH values are <10 kJ/mol, i.e. indication of physisorption. Please explain. In addition, affinity is not a mechanism indication factor, especially when adsorption is not dominant, and high pollutant’s initial concentration leads in low values. Figure 4: Please explain why Figures 4 A, B and C are included in the abstract since the same data are displayed on Figures 4 D, E and F. Figure 6: The authors attributed each peak of XRD spectra in various Cd phases. A proper identification requires more peaks and also it is even harder when Cd content is below 1%, regarding the resolution of the instrument (e.g. CMB and CMB-Cd XRD spectra are almost the same). Please explain or rephrase. Page 12, line no. 325: Please explain why the Si-O-Si bond vibration confirmed the forming of the CdSiO3 phase. Table 3: The determination of the hydrogen ion concentration is confusing at this point. Please explain the corresponding results and also the experimental procedure is missing from Materials and Methods section. Page 13, line no. 352: The examined metals are extremely soluble in the applied experimental condition (pH 6), so they may be leached regardless Cd. Please provide a comparison with the blanks, i.e. without Cd.